# *In Vitro* Mass Propagation of Coffee Plants (*Coffea arabica* L. var. Colombia) through Indirect Somatic Embryogenesis

**DOI:** 10.3390/plants12061237

**Published:** 2023-03-08

**Authors:** Consuelo Margarita Avila-Victor, Víctor Manuel Ordaz-Chaparro, Enrique de Jesús Arjona-Suárez, Leobardo Iracheta-Donjuan, Fernando Carlos Gómez-Merino, Alejandrina Robledo-Paz

**Affiliations:** 1Colegio de Postgraduados, Campus Montecillo, Carretera México-Texcoco Km. 36.5, Montecillo, Texcoco C.P. 56264, Estado de México, Mexico; 2Instituto Nacional de Investigaciones Forestales, Agrícolas y Pecuarias, Campo Experimental Rosario Izapa, Carretera Tapachula-Cacahoatán Km. 18, Tuxtla Chico C.P. 30870, Chiapas, Mexico

**Keywords:** coffee, somatic embryos, *in vitro* propagation, tissue culture

## Abstract

*Coffea arabica* is one of the two most consumed coffee species in the world. Micropropagation through somatic embryogenesis has allowed the large-scale propagation of different coffee varieties. However, the regeneration of plants using this technique depends on the genotype. This study aimed to develop a protocol for the regeneration of *C. arabica* L. var. Colombia by somatic embryogenesis for its mass propagation. Foliar explants were cultured on Murashige and Skoog (MS) supplemented with different concentrations of 2,4-dichlorophenoxyacetic acid (2,4-D), 6-benzylaminopurine (BAP), and phytagel for inducing somatic embryogenesis. In total, 90% of the explants formed embryogenic calli with a culture medium containing 2 mg L^−1^ of 2,4-D, 0.2 mg L^−1^ BAP, and 2.3 g L^−1^ phytagel. The highest number of embryos per gram of callus (118.74) was obtained in a culture medium containing 0.5 mg L^−1^ 2,4-D, 1.1 mg L^−1^ BAP, and 5.0 g L^−1^ phytagel. In total, 51% of the globular embryos reached the cotyledonary stage when they were cultured on the growth medium. This medium contained 0.25 mg L^−1^ BAP, 0.25 mg L^−1^ indoleacetic acid (IAA), and 5.0 g L^−1^ of phytagel. The mixture of vermiculite:perlite (3:1) allowed 21% of embryos to become plants.

## 1. Introduction

Coffee is one of the most important crops worldwide. Coffee species grow positively between 800 to 2000 m above sea level in equatorial and tropical climates [1]. Coffee is native to Africa, and belongs to the Rubeaceae family [2]. Global production of coffee achieved 10.5 million t, while in Mexico it was 240,000 t [3]. The coffee species arabica (*Coffea arabica*) and robusta (*Coffea canephora* var. Robusta) account for 70% and 30% of global production, respectively [4,5,6]. Mexico has sustained a leading position in the list of the most important countries for the production and exportation of coffee. However, Mexico has decreased in the ranking of coffee production and exportation (ranking thirteenth and twelfth, respectively) to date [7]. Several factors have impacted coffee plantations, such as low prices in the world market, old plantations, and lack of financing and investment for culture renovation. Climate change (increasing temperature), poor soil nutrition, and diseases (highlighting coffee rust caused by the fungus *Hemileia vastatrix* Berk. and Br.) have also negatively affected coffee production in Mexico [8,9,10]. Coffee rust is a foliar disease, considered the most devastating for the crop, which in Mexico caused a decrease from 270,000 t to 132,000 t in 2016 [11].

A Comprehensive Coffee Care Plan (CCCP) has been implemented to increase coffee production in Mexico [12]. To achieve this increase, 300,000 ha of plantations must be renewed, demanding approximately 459 million coffee plants within the next few years [11]. As an alternative to increasing coffee production, sustainable, profitable, and environmentally friendly agriculture has been improved, with culture and renewal of coffee plantations with resistant varieties to pests or diseases instead [11]. Colombia variety has shown rust resistance and high tastiness and grain quality standards, low bearing, high production rates, phenotypic uniformity, adaptability, and good yield. Although this variety is not the most cultivated in Mexico, due to its characteristics, it may potentially improve coffee production [9].

Coffee can be propagated sexually (seed) and asexually (cuttings, grafts). The sexual method has a low capacity for multiplication and requires large cultivation areas and long periods for mass propagation. On the other hand, asexual propagation is a time-consuming, laborious, and intensive activity, making it expensive [13]. Biotechnological tools such as micropropagation allow the production of plants of selected genotypes on a large scale, at any time of the year, and in less space. Micropropagation can be carried out by axillary shoot culture, organogenesis, or somatic embryogenesis. The advantage of somatic embryogenesis over organogenesis is that a rooting stage is not required, since the individuals obtained have stem and root meristems; particularly, indirect somatic embryogenesis (previous callus stage) can generate large numbers of plants from few cells [14,15].

Several protocols have been developed to produce somatic embryos in different coffee varieties [10,12,13,14,15,16]; nevertheless, methodologies to induce somatic embryogenesis depend on the plant genotype. In this regard, López-Gómez et al. [16] cultivated leaf explants of 10 genotypes of *C. canephora* and 3 of *C. arabica*, on a culture medium containing 1.1 mg L^−1^ of BAP and 0.5 mg L^−1^ of 2,4-D, to induce somatic embryogenesis. The authors found that only four *C. canephora* genotypes (00-28, 95-8, 97-18, 97-20) and two *C. arabica* genotypes (2000-1128, 2000-692) were able to differentiate embryos. Likewise, Ahmed et al. (2013) tested different combinations of 2,4-D (1.0–2.0 mg L^−1^) and BAP (0.5−3.0 mg L^−1^) to promote the formation of somatic embryos from the foliar explants of the hybrid “Ababuna” of *C. arabica*, finding that a higher proportion of BAP, compared to 2,4-D, promoted the best response.

On the other hand, various authors consider that both growth regulators and stress are the factors that can make differentiated cells become competent to form somatic embryos [17,18,19]. Cells’ response to stress will depend on their physiological state and the level of stress. When stress is extreme, cells can die, but if not, metabolic reprogramming occurs that allows them to adapt, survive, and form embryos [17,20].

The present study aimed to develop a protocol for the regeneration of plants of *C. arabica* L. var. Colombia by somatic embryogenesis for large-scale multiplication, to increase the number of coffee plants, and provide the plants needed for renewing the old plantations, hence maximizing coffee production in Mexico. The working hypothesis was that a greater proportion of cytokinins with respect to auxins, in combination with water stress caused by high concentrations of phytagel, would allow inducing indirect somatic embryogenesis from the foliar explants of the Colombia variety.

## 2. Results

### 2.1. Induction of Embryogenic Callus

Except for the T4 treatment (2,4-D, 1.0 mg L^−1^; BAP, 1.0 mg L^−1^, phytagel, 5.0 g L^−1^), all the treatments tested promoted callus formation in more than 75% of the explants (Figure 1a), while the highest amount of calluses was observed in the explants grown in the combination of 2.0 mg L^−1^ 2,4-D, 0.2 mg L^−1^ BAP, and 2.3 g L^−1^ phytagel (T5) (Table 1).

### 2.2. Embryogenic Calli Production

It was possible to observe significant differences due to the effect of the position of the flask and the subculture time of the calli of the cultivar Colombia. In this way, the calli grown in the flasks (125 mL) with inclination and without subculturing for three months increased their biomass by 18.9 times (18.9 g), the maximum fresh weight accumulated in the different treatments tested (Table 2; Figure 1b). In contrast, calli that were grown upright and not subcultured, only increased their weight by five times (Table 2).

### 2.3. Effect of Embryogenic Callus Induction Media on Differentiation of Somatic Embryos of C. arabica L. var. Colombia

It was possible to observe significant differences between treatments for the number of embryos per gram of fresh callus weight. The maximum value for this variable was obtained in the calli that came from the induction medium that contained 0.5 mg L^−1^ 2,4-D, 1.1 mg L^−1^ BAP, and 5.0 g L^−1^ phytagel (T2), while the lowest number of embryos was observed in the T3 treatment (1.0 mg L^−1^ 2,4-D, 1.0 mg L^−1^ BAP, and 2.3 g L^−1^ phytagel) (Table 3; Figure 1c). 

### 2.4. Development and Maturation

The maturation medium containing 0.25 mg L^−1^ BAP and 0.25 mg L^−1^ IAA (MD2) allowed a greater number of globular embryos to reach the cotyledonary stage (Table 4; Figure 1d), while the percentage of embryos in the initial torpedo stage that reached the cotyledonary stage was not significantly different in any of the tested media (Table 4).

### 2.5. Conversion

The conversion percentage of the embryos into plants was significantly higher when they were placed in the vermiculite:perlite substrate mixture (21.5%) than when they were cultivated in a semi-solid culture medium (7.2%) (Figure 1e). The plants obtained in the semi-solid medium were smaller than those regenerated in the mixture of substrates; in addition, they were more easily acclimatized to *ex vitro* conditions (Figure 1f). Likewise, unlike the roots formed *in vitro*, those that were generated in the substrates were branched and had a greater amount of root hairs.

### 2.6. Histological Analysis of Somatic Embryogenesis

After 30 days of culture in the callus induction medium, the explants began to form calluses in the periphery and mainly in the midrib (Figure 2a). The callus was basically made up of large parenchyma cells, with vacuoles, sparsely evident nuclei, and abundant intercellular spaces. During the following 30 d, the callus cells underwent continuous divisions that allowed their biomass to increase. Then, the primary calli were transferred to the embryo differentiation medium, and small cells began to form on their surface, with dense cytoplasm, evident nuclei, and few intercellular spaces, similar to meristematic cells, which showed a certain degree of organization (proembryogenic masses). The proembryogenic masses formed began to accumulate phenolic compounds in high concentration, after 30 d of culture on the differentiation medium; the presence of said compounds caused the darkening of said masses (Figure 2a) and only those that produced phenols managed to form embryos. At 120 d of culture in the differentiation medium, embryos in the globular stage were formed from the proembryogenic masses (Figure 2b); these embryos showed a protodermis that covered the periphery of the embryo and that was made up of a series of small cells with an evident nucleus (Figure 2c). When the globular and heart embryos (Figure 2c,d) were cultured in the maturation medium, they reached the torpedo (Figure 2e,f) and cotyledonary stage after 30 and 45 d, respectively. In the torpedo and cotyledonary stage embryos, it was possible to observe a group of elongated cells with visible nuclei that were distributed throughout the embryo and that constituted the vascular procambium zone (Figure 2f).

## 3. Discussion

### 3.1. Induction of Embryogenic Callus

Even though all treatments tested induced callus formation in the explants of var. Colombia, after 15–30 d of culture, the medium containing 2 mg L^−1^ 2,4-D, 0.2 mg L^−1^ BAP, and 2.3 mg L^−1^ phytagel not only promoted that a high percentage of explants formed calluses but also formed calluses in greater quantities. In contrast, Gatica-Arias et al. [22] observed that 87% and 67% of the explants of the Caturra and Catuaí varieties of *C. arabica*, respectively, formed calluses in the presence of 4 mg L^−1^ kinetin, 1 mg L^−1^ 2,4-D, and 2.0 g L^−1^ Gelrite, after four months, that is, in a higher ratio of cytokinin to auxin. Likewise, Ahmed et al. [13] found that the best response in terms of callus formation (96%) was achieved when the explants of the hybrid “Ababuna” (*C. arabica*) were grown in an MS medium added with 1.0 mg L^−1^ 2, 4-D and 2.0 mg L^−1^ BAP for 90 d.

Likewise, the phytagel is a heteropolysaccharide used in the culture medium as a gelling agent. This compound is generally supplemented at a concentration ranging between 1.25–2.5 g L^−1^. The gelling agent is the matric component of the water potential in the medium, that influences the amount of available water (the higher the concentration, the greater the retention) and the mobility of the solutes toward the cells of the explant, which will be reflected in their growth [23].

The fact that in the culture media tested containing a lower concentration of phytagel, a greater amount of calluses was formed than in those with a higher concentration (5.0 g L^−1^), indicates that cell growth and division occurs more efficiently when there is more free water. In this regard, Meneses et al. [24] observed that the embryogenic calli of indica rice (*Oryza sativa*) grew better when the concentration of phytagel was less than 2.4 g L^−1^.

### 3.2. Embryogenic Calli Production

The position and frequency of subcultures did have a significant effect, registering the highest biomass accumulation (18.9 times) when the flasks were kept tilted and no subcultures were carried out. In this regard, Gatica-Arias et al. [22] obtained 1.4 g of fresh callus weight of *C. arabica* L. cv. Catuaí from 250 mg of fresh callus weight (5.6 times) grown in 125 mL Erlenmeyer flasks with 25 mL of liquid medium, for four weeks in darkness. Likewise, de Rezende et al. [25] observed that the calli of Clone 3 [Siriema (*C. racemose* × *C. arabica*)] grown in MS salts at 50%, 1.1 mg L^−1^ 2,4-D, 1.0 mg L^−1^ indole butyric acid (IBA), and 2.0 mg L^−1^ 2-isopentenyladenine (2iP) for 63 days (with a subculture every 21 days), increased their fresh weight by 7 times, while those of Clone 28 (cross of Red Catuaí IAC 44 × Timor Hybrid CIFC 2750) increased their biomass by 9.8 times, under the same growing conditions.

In general, when cells remain in suspension, the absorption and utilization of nutrients, growth regulators, and water are greater than when they are cultured in semi-solid media [22]. However, in a liquid culture, anaerobic conditions can occur because oxygen diffuses less easily, which causes the oxygen level in the tissue to be reduced and, thus, the accumulation of biomass is less [26]. In the present investigation, when the flasks remained tilted, the cells were only temporarily submerged in the culture medium, which could have allowed the oxygen present in the flask to be more available to them, promoting their metabolism and growth to occur optimally [27].

The results obtained show that the nutrients in the culture medium can support the growth of calluses for three months without the need for it to be renewed, which allows for saving inputs and labor, reducing the risk of contamination and the probability of somaclonal variation occurring [28]. Keeping the flasks in a tilted position and without frequent subcultures can be an alternative for the successful multiplication of coffee embryogenic cultures.

### 3.3. Differentiation (Formation) of Somatic Embryos

The results showed that the presence of 2,4-D and BAP in a 1:2 ratio and the highest concentration of phytagel (5.0 g L^−1^) in the culture medium had a synergistic effect to induce somatic embryogenesis in the calluses of cv. Colombia, allowing the highest number of embryos to be formed (118.7 embryos per gram of callus, before the multiplication stage). In this regard, Ahmed et al. [13] obtained the maximum number of embryos (58 embryos per magenta) from the calli of the hybrid “Ababuna” of *C. arabica*, generated in a medium that contained 1 mg L^−1^ 2,4-D and 2 mg L^−1^ BAP, which were then cultured in the presence of 4 mg L^−1^ BAP for six months. 

Some authors [17,18,19,29] point out that the induction of somatic embryogenesis is the result of a combination between growth regulators and stress that can be caused by the removal of cells from the explant of the mother plant, as well as the conditions of *in vitro* cultures (high and low concentration of salts and gelling agent, osmotic shock, temperature, high or low light intensity, water stress, among others) [17,18,19]. However, it has been found that the endogenous hormone level is highly variable in the competent cells of different genotypes or not significantly different between a competent and a non-competent cell [17]. For this reason, the induction of somatic embryogenesis could not be attributed solely to growth regulators, but stress is considered a fundamental factor in this process. It has been suggested that somatic embryogenesis is a response of cells to extreme stress because in the early stages of embryo formation, the expression of different genes related to stress, oxidative stress, and cell division has been detected [17,30].

Potters et al. [31] point out that regardless of the type of stress to which cells are exposed, their division is stimulated, their growth is inhibited, and changes occur in their state of differentiation. Cells are brought from their differentiated condition to adapt to their new environment, leading to genetic and metabolic reprogramming. Under certain conditions, cells ensure their survival, leading to the formation of embryos [29].

On the other hand, 2,4-D has been used to induce somatic embryogenesis in more than 60% of the protocols [32]. It is suggested that 2,4-D, above certain concentrations, can act as a growth regulator or as an agent that generates stress. 2,4-D can modify the endogenous metabolism of IAA by affecting its binding to auxin-binding proteins [18]. Likewise, it is suggested that this regulator generates hypermethylation, which maintains cells in a state of high mitotic activity and in a proembryogenic phase [24].

Moreover, while callus induction using 2.3 g L^−1^ phytagel promoted the best response, including 5.0 g L^−1^ in the differentiation medium generated the highest number of embryos per gram of callus. This response could be because the phytagel, by acting as a matrix, decreases the availability of water and nutrients from the culture medium for the cells and at high concentrations causes water stress [33].

### 3.4. Development, Maturation, and Conversion

During maturation, somatic embryos undergo biochemical (accumulation of reserve substances) and morphological changes (increase in size) which are essential for their subsequent development, germination, and conversion into photoautotrophic plants [34].

The results of the present investigation show that including auxins and cytokinins in the development medium in a 1:1 ratio, allowed embryos in the globular or torpedo stage to pass to the cotyledon stage and that low concentrations (0.25 mg L^−1^) of both BAP and IAA (MD2 medium) were sufficient to induce this response. Likewise, it was observed that embryos in the globular stage can reach the cotyledonary stage when cultured in a single medium and that it is not necessary to separate the embryos in the different stages (globular and torpedo) to promote their development since the MD2 medium works for both; this allows for reducing time, supplies, and labor.

On the other hand, different authors point out that the success in the conversion depends to a great extent on how efficient the maturation of the embryos was, as those that do not mature show low germination rates [35]. For the Colombia variety, the conversion of the embryos was higher when cultured in a mixture of perlite and vermiculite than in agar; however, the conversion rate was not greater than 25%. These results coincide with those obtained by Georget et al. [36], who found that the average conversion rate of somatic embryos to seedlings of hybrids H1 (T5296 × RS), H3 (Caturra × ET 531), H10 (T5296 × RS), H16 (T5296 × ET01A1), H17 (Catuai × ET59A2), and other *C. arabica* hybrids in a substrate (peatmoss and sand) was only 37%. The embryos of the Colombia variety grown in the mixture of substrates resulted in larger plants with a greater capacity to adapt to *ex vitro* conditions than those that remained on agar; it could be because the physical characteristics of the substrates, such as porosity, allow a better oxygen and water balance, facilitating the diffusion of nutrients and the formation of more vigorous roots [37,38]. The aforementioned indicates that germination and conversion are still a bottleneck for coffee micropropagation through somatic embryogenesis [35,36].

### 3.5. Histological Analysis of Somatic Embryogenesis

The histological analysis of the cultures allowed us to observe that the primary callus was made up of cells that were large, elongated, with vacuoles and large intercellular spaces as described by Bartos et al. [39] in *C. arabica* cv. Catuaí Vermelho. After the primary callus was transferred to the differentiation medium, proembryogenic masses began to form; these masses had a certain degree of organization in which the somatic proembryos arose, as was also observed in the *C. arabica* Caturra and Catuaí varieties [22,39,40].

The fact that only calli that produced phenols formed somatic embryos is consistent with the observations of Martins et al. [41], who found that the presence of phenols in the calli of *Arbutus unedo* and *A. canariensis* was a prerequisite for the formation of somatic embryos. Tonietto et al. [42] indicate that during the early stages of somatic embryogenesis, a large number of reactive oxygen substances (ROS) is generated; to defend against ROS, cells develop different enzymatic and non-enzymatic antioxidants (tocopherols, carotenoids, phenolic compounds, among others), systems that catalyze the breakdown of these oxygen molecules [43].

It was possible to observe the transition of the globular embryos to the heart, torpedo, and cotyledon phase in the following 45 days of culture in the development and maturation medium. Globular stage embryos showed an outer layer of cells covering the embryo, the protodermis, as observed by Bartos et al. [39] in the somatic embryos of *C. arabica* var. Caturra. In the torpedo stage, procambium cells connected to the stem and root meristem in embryos of the Colombia variety. In this regard, Nic-Can et al. [44] consider that the presence of the polarized fundamental meristems with differentiating procambium cells are the main anatomical characteristics of the torpedo somatic embryos of *C. canephora*. Likewise, the lack of vascular connection between the regenerated somatic embryos and the original explant was evident.

The histological analysis of the cultures obtained in the present investigation allowed us to confirm the embryogenic nature of the regenerated plants. These structures and cellular characteristics could be used as morphological markers to detect those cells that have embryogenic competence, as well as to define with more precision, the ideal time to transfer the cultures to the following stages, thereby contributing to reducing the time to obtain somatic embryos and the optimization of micropropagation protocols [39,44].

Coffee is the second-most consumed beverage in the world; *C. arabica* represents 70% of the production of this crop worldwide, which makes it the most important species of the genus. Even though there are protocols for somatic embryogenesis in different species and varieties of coffee [22,36,45,46], for *C*. *arabica* var. Colombia, the information is limited [47]. More research is required to establish an efficient and reproducible protocol for the propagation of plants of this variety, which in addition to being tolerant to coffee rust, has outstanding agronomic characteristics and wide adaptation to the environments where coffee is grown, as well as having a good taste quality [48]. Having an efficient micropropagation protocol for the var. Colombia will allow to have enough plants to establish and renew plantations with which economic and environmental costs are reduced, by making less use of agrochemicals (fungicides) to combat the fungus that causes rust of the coffee.

## 4. Materials and Methods

### 4.1. Disinfection

The first and second pairs of young leaves, healthy and undamaged, were obtained from one-year-old plants grown in a greenhouse at 32 ± 4 °C, 400 μmol m^−2^ s^−1^ of light intensity, with a relative humidity of 40%. The plants were watered every third day, fertilized every four months with Multicote 15-7-15^®^ (slow-release fertilizer, Green Forest, Puebla, México), and fumigated monthly with Cupravit^®^ (copper oxychloride, Bayer^®^, Querétaro, México). Then, leaves were disinfected by immersion in 0.1% fungicidal solution (Promyl^®^, Promotora Técnica Industrial, S.A. de C.V., Morelos, México): methyl-1-butyl carbamoyl-2-benzimidazole carbamate) for 15 min and then rinsed with sterilized distilled water. After that, leaves were placed in sodium hypochlorite (1.2% active chlorine) for 20 min and then rinsed with sterilized distilled water.

### 4.2. Induction of Embryogenic Callus

Under sterile conditions in the laminar air flow, segments of 1 cm^2^ (explants) were obtained from disinfected leaves and placed in 90 × 15 mm Petri dishes containing 30 mL of MS (Murashige and Skoog) basal medium [49]. The media were supplemented by combining several concentrations of two growth regulators and two concentrations of phytagel (Table 5). Culture media were enriched with 200 mg L^−1^ of ascorbic acid, 100 mg L^−1^ of citric acid (antioxidants), and 30 g L^−1^ of sucrose; pH was adjusted to 5.7 and then, media were sterilized in an autoclave at 121 °C for 20 min.

The cultures were incubated in a growth chamber under dark conditions at 26 ± 2 °C. After two months, the following were evaluated: (a) the percentage of explants that generated calluses and (b) the amount of calluses that formed on each explant (explant area covered with calluses). A scale to evaluate the amount of calluses was used, considering the percentage of area covered with calluses as follows: 1: 0–20%; 2: 21–40%; 3: 41–60%; 4: 61–80%; 5: 81–100%. The experiment for callus induction had a factorial arrangement with two factors: growth regulator combinations (3), and phytagel concentration (2), resulting in six treatments. Each treatment was reproduced in ten repetitions, each (Petri dish) containing six explants.

### 4.3. Embryogenic Calli Production

Five hundred milligrams were placed in 125 mL flasks with 50 mL of full MS medium supplemented with 1 mg L^−1^ 2,4-D, 1 mg L^−1^ BAP, and 30 g L^−1^ sucrose; the flasks were shaken (orbital shaker, Thomas Scientific^®^, Swedesboro, NJ, USA) at 110 rpm in darkness.

To evaluate the increase in biomass of the calli, three conditions were tested: the first consisted of cultivating the calli in 125 mL flasks and keeping them upright in an orbital shaker, performing subcultures every four weeks for three months. In the second and third conditions, the calli were cultured in 125 mL flasks without performing subcultures; after three months the weight of the calluses was recorded. The difference between the second and third conditions consisted of placing the flasks on the shaker in a vertical or inclined position (30° approximately), respectively.

A completely randomized experimental design with three treatments and ten repetitions was used; one flask was considered as a repetition.

### 4.4. Differentiation (Formation) of Somatic Embryos

After 60 d, the embryogenic callus formed in each of the induction media was separated and placed in Petri dishes of 60 × 15 mm with 15 mL of differentiation medium, which contained the basal salts of Yasuda et al., modified [21], 1.1 mg L^−1^ BAP, 5.0 g L^−1^ phytagel, and 30 g L^−1^ sucrose. The cultures were incubated at 26 ± 2 °C in darkness and after four months, the number of embryos per gram of callus was quantified.

The experiment had a factorial arrangement with two factors: growth regulator combinations in the induction medium (3), and the concentration of phytagel (2), giving rise to six treatments; each treatment consisted of ten repetitions and one repetition was made up of a Petri dish with calluses (500 mg).

### 4.5. Development and Maturation of Embryos

The embryos in the globular state, heart, and initial torpedo were placed in flasks of 30 mL with 10 mL of two culture media made with the basal salts of Yasuda et al., modified [21], 5.0 g L^−1^ phytagel, 30 g L^−1^ sucrose plus 0.5 mg L^−1^ 6-furfurylaminopurine (kinetin), and 0.5 mg L^−1^ indoleacetic acid (IAA) (MD1 medium) or 0.25 mg L^−1^ BAP and 0.25 mg L^−1^ IAA (medium MD2). The number of cotyledonary stage embryos was evaluated after six weeks. A completely randomized experimental design with two treatments was used. Each treatment (culture medium) consisted of 15 repetitions, considering as an experimental unit a flask with 10 embryos.

### 4.6. Conversion

The cotyledonary stage embryos were transferred to 250 mL flasks containing 30 mL with half MS medium, 2.3 g L^−1^ phytagel, and 20 g L^−1^ sucrose or a mixture of vermiculite: perlite (3:1, particle size 0.5 mm); the mixture was moistened with 30 mL of the previously described medium without agar. The cultures were kept at 26 ± 2 °C and 16 h of fluorescent cold white light (60 µmol m^−2^ s^−1^). After two months, the number of embryos that became plants was counted. The experimental design was a completely randomized one with two treatments and ten repetitions; a flask with ten embryos was considered as an experimental unit.

### 4.7. Histology

Embryogenic calli at different stages of development were fixed in a mixture of 10% formaldehyde, 5% acetic acid, 52% ethanol, and 33% water (*v*/*v*); then, they were dehydrated with a series of ethanols (30, 40, 50, 70, 85, 100%) and xylenes (ethyl alcohol 50% -xylene 50%, pure xylene) and embedded in paraffin. Thick sections of 10 µm were made with a rotary microtome (American Optical^®^, model Spencer 820, Vernon Hills, IL, USA). The samples were treated with xylene (100%), and a series of alcohols (ethanol 50, 70, 85, 100%) to eliminate paraffin; then, they were stained with fixed O-safranin-green and infiltrated and embedded in synthetic resin [50]. The samples were observed with the help of a Carl Zeiss^®^ (Oberkochen, Germany) optical microscope (Tessovar model) and the images were captured with a Paxcam^®^ (Villa Park, IL, USA) digital camera.

### 4.8. Statistical Analysis

The data were statistically processed using an analysis of variance (ANOVA); the means of the treatments were compared utilizing a Tukey test with a significance level of 5%. The statistical program SAS version 9.0 was used.

## 5. Conclusions

The culture conditions established in the present investigation allowed the development of a protocol by which it is possible to obtain *in vitro* plants of *C. arabica* var. Colombia, through indirect somatic embryogenesis. The osmotic stress generated by the high concentration of phytagel in the medium increased the ability of the explants to form embryos. The low concentration of nutrients in the callus multiplication medium and the temporary contact of the cells with the culture medium made it possible to multiply them efficiently. The regenerated embryos were able to grow into plants and establish themselves under *ex vitro* conditions.

## Figures and Tables

**Figure 1 plants-12-01237-f001:**
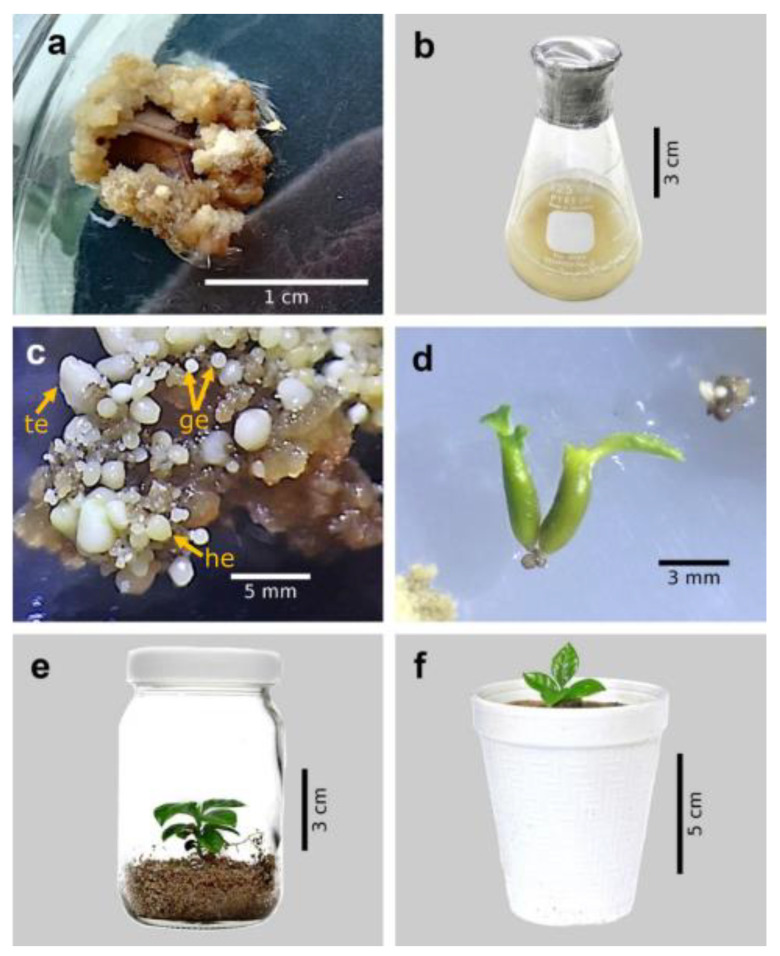
Somatic embryogenesis in coffee (*Coffea arabica* L. var. Colombia). (**a**) Embryogenic callus from leaf explants after eight weeks. (**b**) Embryogenic callus production in liquid medium after 12 weeks. (**c**) Somatic embryos in different stages of development after 150 d of culture differentiation medium. (**d**) Cotyledonary embryos after four weeks of culture on maturation medium. (**e**) *In vitro* plant grown in vermiculite:perlite mixture (3:1). (**f**) *In vitro* regenerated plant grown under greenhouse conditions. ge: globular, he: heart, te: torpedo.

**Figure 2 plants-12-01237-f002:**
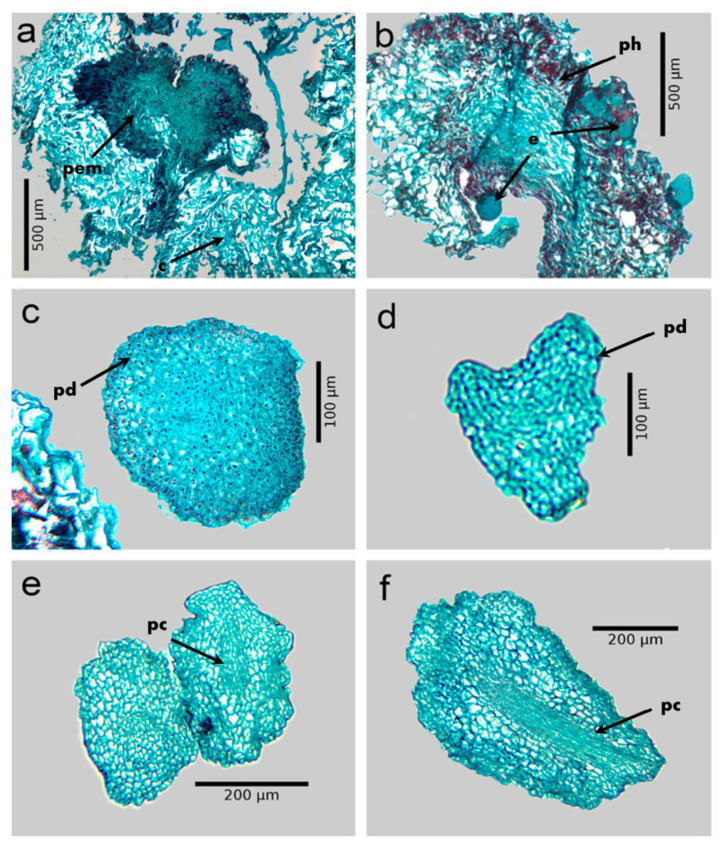
Histology of somatic embryogenesis of *Coffea arabica* L. var. Colombia. (**a**) Proembryogenic mass forming from a primary callus. (**b**) Somatic embryos differentiated on the proembryogenic mass that produces phenols. (**c**) Globular embryo at 120 d of culture in the differentiation medium. (**d**) Heart stage embryo after 135 d of culture in differentiation medium. (**e**) Early torpedo embryo. (**f**) Torpedo embryo after 150 d of culture on the differentiation medium, showing protodermis and procambium. c: primary callus; pem: proembryogenic mass; ph: phenols; e: embryos; pd: protodermis; pc: procambium.

**Table 1 plants-12-01237-t001:** Effect of different combinations of 2,4- D and BAP at two concentrations of phytagel on induction of embryogenic callus from leaf explants of *C. arabica* var. Colombia after two months of culture. All cultures were kept under dark conditions at 26 ± 2 °C.

Treatment	Growth Regulators (mg L^−1^)		Phytagel (g L^−1^)	Explants with Callus (%)	† Callus Quantity
	2,4-D	BAP			
1	0.5	1.1	2.3	90.00 a	1.03 bc
2	0.5	1.1	5.0	75.74 ab	1.00 c
3	1.0	1.0	2.3	90.00 a	1.20 b
4	1.0	1.0	5.0	69.00 b	1.15 bc
5	2.0	0.2	2.3	90.00 a	1.38 a
6	2.0	0.2	5.0	85.05 a	1.03 bc

† Explant area covered with calluses: 1: 0–20%; 2: 21–40%; 3: 41–60%; 4: 61–80%; 5: 81–100%. Means with the same letter are not statistically different (Tukey α = 0.05).

**Table 2 plants-12-01237-t002:** Effect of flask position and subculture on biomass production of embryogenic calli of *C. arabica* L. var. Colombia, after three months.

Flask Position and Subculture	Biomass (g)
Inclination—without subculture	18.94 a
Vertical—without subculture	5.17 b
Vertical—with subculture	1.27 b

Means with the same letter are not statistically different (Tukey α = 0.05).

**Table 3 plants-12-01237-t003:** Number of embryos formed from calli of *C. arabica* var. Colombia obtained in 6 different induction media, after 17 weeks cultured in a differentiation medium.

Treatment	Growth Regulators (mg L^−1^)		Phytagel (g L^−1^)	Embryos per Gram
	2,4-D	BAP		
1	0.5	1.1	2.3	44.03 b
2	0.5	1.1	5.0	118.74 a
3	1.0	1.0	2.3	7.81 c
4	1.0	1.0	5.0	27.94 b
5	2.0	0.2	2.3	23.75 b
6	2.0	0.2	5.0	26.19 b

Means with the same letter are not statistically different (Tukey α = 0.05). Differentiation medium: modified Yasuda et al. [21] basal salts, 1.1 mg L^−1^ BAP, 5.0 g L^−1^ phytagel, 30 g L^−1^ sucrose.

**Table 4 plants-12-01237-t004:** Percentage of globular and torpedo embryos of *Coffea arabica* var. Colombia that reached maturity (cotyledonary stage) after 30 days of culture on the maturation media.

Culture Medium	Globular Embryos	Torpedo Embryos
	Maturation (%)	Maturation (%)
MD1	39.24 b	40.90 a
MD2	51.72 a	45.41 a

MD1: Kinetin, 0.5 mg L^−1^; IAA, 0.5 mg L^−1^; MD2: BAP, 0.25 mg L^−1^; IAA, 0.25 mg L^−1^. Means with the same letter are not statistically different (Tukey α = 0.05).

**Table 5 plants-12-01237-t005:** Treatments used to induce the embryogenic calli from leaf explants of *C. arabica* var. Colombia.

Treatment	Growth Regulators (mg L^−1^)		Phytagel (g L^−1^)
	2,4-D	BAP	
T1	0.5	1.1	2.3
T2	0.5	1.1	5.0
T3	1.0	1.0	2.3
T4	1.0	1.0	5.0
T5	2.0	0.20	2.3
T6	2.0	0.20	5.0

2,4-D, 2,4-dichlorophenoxyacetic acid; BAP, 6-benzylaminopurine.

## Data Availability

All data generated in this study are included in the tables and figures.

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
