# Peer review of "In Vitro Mass Propagation of Coffee Plants (Coffea arabica L. var. Colombia) through Indirect Somatic Embryogenesis"

_plants, 2023, doi:10.3390/plants12061237_

Round 1
Reviewer 1 Report
Coffee is a crop of great economic importance. Development of an efficient method of micropropagation seems to be important for more efficient production of seedlings. The authors developed a method of propagation of the Colombia variety of coffee. I believe that including a few different genotypes in the investigation would bring more value to the study, because of the high dependence of the genotype on the somatic embryogenesis process.
The manuscript has been written with great care, and all its chapters are coherent and logical. The methodology chapter lacks the characteristics of donor plants and their growth conditions, which may be essential for the obtained results. I also wonder why the Phytagel concentrations of 2.3 g L-1 or 5.0 g L-1 were selected, since in the discussion the authors clearly state that in previous studies concentrations of up to 2.4 g L-1 are used. Similarly, please explain why exactly such concentrations of growth regulators were used in the inducing medium. I also don't see the logic in the subsection “Embryogenic callus multiplication”. The reduction of flask volume with the proportionally lower weight of the fresh callus could not have had an effect on the results.
The description of obtained results, photographic and tabular documentation, as well as the discussion of results are very carefully prepared and supported by literature. Conclusions logically follow the results obtained.
Reviewer 2 Report
Production of coffee plants (Coffea arabica L. var. Colombia) by indirect somatic embryogenesis
I have some comments:
The language of the MS should be edited and improved by native English language
The title: The authors should use specific expressions related to plant tissue culture, where the word production not suitable here
The study aimed to develop a protocol for the regeneration of plants of Coffe arabica L. var. Colombia by somatic embryogenesis for mass or large scale multiplication
So, it would be better if the title change to:
In Vitro Mass Propagation of Coffee Plants (Coffea arabica L. var. Colombia) through Indirect Somatic Embryogenesis
Please, check the affiliation, and you should write the name of country for authors number 2 Chiapas. C. P. 30870, México Chiapas is a southernmost state in México is it right?
Introduction:
You should mention the weather condition suitable for the production of coffee and what is the origin of coffee plant, which plant family coffee is belong to.
What is the global production of coffee in general (Ton) and in México In particular
Line 34: What do mean (bad weather conditions)? Refer to them in the introduction
Is Coffea arabica L. var. Colombia the most common cultivated coffee in México? That is why you interested in its propagation for increasing the production, please clarify that.
You should refer to the conventional (traditional) methods of propagation of coffee and the problems that face these methods and why we have to use tissue culture techniques for its propagation
By which methods Coffee plant could be propagated in vitro and why somatic embryogenesis is promising?
The last paragraph of introduction you have to focus mainly on the aim of study
The present study aimed to develop a protocol for the regeneration of plants of C. arabica L. var. Colombia by somatic embryogenesis for large scale multiplication, to increase the number of coffee plants and provide the plants needed for renewing the old plantations, hence maximize the coffee production in Mexico
Finally, you should rewrite the introduction to be contain the previous important information for the reader
Material and Methods:
Disinfection:
What is age of coffee trees when you took the leaf explants? Line 293
Describe the conditions (temperature, light density, relative humidity, irrigation and fertilization, if there is pesticide or fungicide was used, etc) in the greenhouse where the plants (source of explants) were grown (add these information in the text)
Induction of embryogenic callus:
Why you added 200 mg L-1 of ascorbic acid, 100 mg L-1 of citric acid to the medium to induce embryogenic callus from leaf explants (add this information or explanation in the text) Line 302
After immersion the leaves in the fungicidal solution, I think they should be rinse well with sterilized distilled water then you can use sodium hypochlorite (maybe reaction between them happen causing release harmful substance to the explants what do you think? I didn’t use this fungicide before for surface sterilization of explants
Before the following sentences colored in yellow, you should write (Under sterile conditions in the laminar air flow, the excised leaves were disinfected ……………………..
Then, leaves were disinfected by immersion in 0.1% 293 fungicidal solution (Promyl®: methyl-1-butyl carbamoyl-2-benzimidazole carbamate) for 15 minutes. After that, leaves were placed in sodium hypochlorite (1.2% active chlorine) for 20 minutes and then rinsed with sterilized distilled water.
The fungicidal solution (Promyl®) which company and country produced it
Table (5): Treatments used to induce the embryogenic calli from leaf explants of C. arabica var. Colombia.
Could you present the table in a good manner? As shown to be easy to understand
|
Treatment |
Growth regulators (mg L-1) 2,4-D BAP |
Phytagel (g L-1) |
|
|
T1
T2 |
0.5
0.5
|
1.1
1.1
|
2.3
5.0
|
|
|
|
|
|
Line 306:
The cultures were incubated in a growth chamber under dark conditions at 26 ± 2 °C
Line 307: the amount of callus that formed each explant what do you mean?
The amount of callus that formed from each explant
Embryogenic Calli Multiplication: it is better to use Embryogenic Calli production
We use the word multiplication mainly for shoots
Line 315: One gram of embryogenic callus 100 % MS (you can write full MS) it is better
Line 318: (orbital shaker, Thomas Scientific®) which company and country?
Why did you use T4 (1 mg L-1 2, 4-D +1 mg L-1 BAP + phytagel, 5.0 g L-1), that gave the lowest value of callus induction, for Embryogenic Calli production explain, please
Why you didn’t use T5 (the best treatment for callus induction)
One gram of callus was placed in 250 mL flasks with 100 mL
0.5 grams were placed in 125 mL flasks with 50 mL of the same medium
The same proportional between the amount of calls, volume of flask and volume of the medium. I think it should be not consider as a factor in the experiment and you have to study the effect of the three different incubation conditions only on calli production
Differentiation (formation) of somatic embryos:
Line 329: The embryogenic callus formed in each of the induction media
Line 337: Petri dish with callus (what is the amount of callus in each petri dish)
Line 349: replace 50% MS medium with half MS medium it is commonly used
Line 351: The cultures remained were kept is better
Results:
The title of the table should contain all details especially the incubation conditions
Table 1. Effect of different combinations of 2,4- D and BAP at two concentrations of phytagel on induction of embryogenic callus from leaf explants of C. arabica var. Colombia after two months of culture. All cultures were kept under dark conditions at 26 ± 2 °C.
We don’t have to return back to section of Materials and Methods
Table 2.
Effect of flask position and subculture on biomass of embryogenic calli of C. arabica L. var. Colombia
Line 71: without subculturing for three months I think the long time of culture might be resulted in instability due to somaclonal variation
Flask-subculture position Flask position and subculture
Figure 1.
Line 80: after sixteen weeks of culture after two months (eight weeks) in M and M
Embryogenic callus multiplication in liquid replace by production
Line 81: Cotiledonary cotyledonary
Line 82: embryos after six weeks of culture four weeks
Vitroplant grown in vitro plant
Vermiculite:perlite mixture (3:1)
2.3 Effect of embryogenic callus induction media on differentiation of somatic embryos of C. arabica L. var. Colombia
Delete the subtitle 2.3.1. Effect of the combination of growth regulators and phytagel interaction
Table 3. Number of embryos formed from calli obtained from foliar explants of Coffea arabica var. Colombia exposed to different treatments.
Number of embryos formed on calli produced from leaf explants of C. arabica var. Colombia and cultured on embryogenic callus induction media
Conclusions
Line 372: ……..is possible to obtain in vitro plants of C. arabica var. Colombia
Lne 373: …….in eight months in twelve months check your experiments
The produced plants have been obtained by indirect somatic embryogenesis (passing through the callus phase), also, the high concentration of 2,4 D, long period of culture can cause somaclonal variation
So, why you didn’t examine the genetic stability of the in vitro raised plants
You know that the main target when we establish the protocol of in vitro propagation is getting true to type plants (identical to the mother plants)
I think it would be better if the protocol depend on direct somatic embryogenesis (which was established before for coffee plant) as shown below and low concentration of 2, 4 D
DIRECT AND INDIRECT SOMATIC EMBRYOGENESIS
ON ARABICA COFFEE (Coffea arabica) Ibrahim et al, 2013
References:
Some of references have been written in other language, they must be written in English
Minor revision

Reviewer 3 Report
The paper presents a standard procedure for modification of already existing protocols for another genotype. There are already protocols for SE in coffee. Therefore, it is necessary to stress in the intruduction what is the hypothesis, what should be improved or explained scientifically. This part is weak. Therefore also the discussion is rather descriptive. Moreover, the description of the experimental design needs revision.
In the current version in my opinion the paper is not ready to be published in this journal. More remarks are annotated to the manuscript.

Round 2
Reviewer 3 Report
The manuscript was improved. There are few yellow marks in the manuscript tht should be considered.
The part with the osmotic effects is still (in my opinion) not sufficient explained. This is a rather new aspect that needs more attention.

Author Response
Reviewer 3: Please see the attachment.
Comment to editor: The document has already been reviewed by a person whose native language is English.
